# Assessment of the Clinical and Economic Impact of Different Immunization Protocols of Measles, Mumps, Rubella and Varicella in Internationally Adopted Children

**DOI:** 10.3390/vaccines8010060

**Published:** 2020-02-01

**Authors:** Sara Boccalini, Angela Bechini, Cecilia Maria Alimenti, Paolo Bonanni, Luisa Galli, Elena Chiappini

**Affiliations:** 1Department of Health Sciences, University of Florence, Viale GB Morgagni 48, 50134 Florence, Italy; angela.bechini@unifi.it (A.B.); cecilia.alimenti@stud.unifi.it (C.M.A.); paolo.bonanni@unifi.it (P.B.); luisa.galli@unifi.it (L.G.); elena.chiappini@unifi.it (E.C.); 2Meyer Children’s University Hospital, Viale Pieraccini 24, 50139 Florence, Italy

**Keywords:** internationally adopted children, vaccination, vaccine preventable diseases, children, adoptees, measles, mumps, rubella, varicella, infectious diseases

## Abstract

The appropriate immunization of internationally adopted children (IAC) is currently under debate and different approaches have been suggested. The aim of this study is to evaluate the clinical and economic impact of different strategies of measles, mumps, rubella, and varicella (MMRV) immunization in IAC in Italy. A decision analysis model was developed to compare three strategies: presumptive immunization, pre-vaccination serotesting and vaccination based on documentation of previous immunization. Main outcomes were the cost of strategy, number of protected IAC, and cost per child protected against MMRV. Moreover, the incremental cost-effectiveness ratio (ICER) was calculated. The strategy currently recommended in Italy (immunize based on documentation) is less expensive. On the other hand, the pre-vaccination serotesting strategy against MMRV together, improves outcomes with a minimum cost increase, compared with the presumptive immunization strategy and compared with the comparator strategy. From a cost-effectiveness point of view, vaccination based on serotesting results in being the most advantageous strategy compared to presumptive vaccination. By applying a chemiluminescent immunoassay test, the serology strategy resulted to be clinically and economically advantageous. Similar results were obtained excluding children aged <1 year for both serology methods. In conclusion, based on our analyses, considering MMRV vaccine, serotesting strategy appears to be the preferred option in IAC.

## 1. Introduction

Appropriate immunization of internationally adopted children (IAC) is currently under debate and different approaches have been suggested [1]. In the literature, three possible approaches to vaccinate IAC have been described [1,2]. 

The first approach is to assess and accept reliable documentation of previous vaccinations and then immunize the child, following the local schedule [3]. In order to assess the reliability of vaccine documentation, physicians should evaluate the dates of vaccine administration, the number of doses administered, intervals between doses, and the IAC age at the time of administration. This is the strategy recommended in Italy. 

The second approach is to test specific antibody titers. Children should be immunised if no protective antibody titers are found. On the other hand, if protective antibody levels are detected, IAC should be provided with vaccinations according to their age and the national schedule [4]. Testing antibody titers could be carried out for many vaccine preventable diseases (VPDs). However, the performance of serologic tests varies according to the considered VPDs. Limited data suggest that this strategy could be cost-effective for older children [5]. This approach reduces the number of unnecessary injections. [6]. 

The third approach is to re-immunise all IAC without considering their vaccination records, as there is no evidence of adverse events from extra doses of most vaccines [7].

It is not currently known which of these three options provides the most cost-effective approach. 

Previous studies found different results depending on the VPDs and the prevalence of the disease in the population [8,9,10,11]. Serology for varicella in refugees older than five years of age was observed to be cost-effective in one study of 637 children [8]. On the other hand, presumptive vaccination was found to be cost-effective for HBV and HAV in two studies when the prevalence was >40% and >22%, respectively [9,10].

A previous economic analysis compared pre-vaccination serotesting and presumptive immunization for poliomyelitis, diphtheria, and tetanus in IAC. The authors concluded that appropriate immunization of IAC is currently under debate and different approaches have been suggested, presumptive immunization for the poliovirus improved outcomes and saved costs compared with serotesting. However, the results for tetanus and diphtheria were less definitive and were influenced by the assumed seroprevalence and compliance to the three-dose-vaccination schedule [11]. To our knowledge, no previous study has investigated the preferred strategy in IAC relative to measles mumps rubella varicella (MMRV) vaccination.

MMRV are highly communicable infectious diseases. Due to inadequate immunization coverage outbreaks of measles and other VPDs have been reported worldwide in recent years among the general population [12]. 

In total, 13,460 cases of measles have been reported in adults and children by the European Country for Disease Prevention and Control between the 1 December 2018 and 30 November 2019. In November, Romania and France counted the highest number of cases, with 79 and 48 respectively [13]. 

In Italy from 1 January 2019 to 31 October 2019, 1605 cases of measles and 19 cases of rubella were reported [14]. The number of measles cases decreased compared to the previous year, in fact outbreaks were reported in 2017 and 2018. In fact, from the 1st of January to the 30th November 2018, 2427 cases of measles were notified [15] and during 2017 (between 1 January and 10 December) 4885 cases of measles were notified in Italy [16].

Individuals at the highest risk for MMRV infections include unvaccinated or inadequately vaccinated persons. The control of MMRV is ensured when reaching the threshold of herd immunity (95% of vaccination coverage with 2 doses) [17]. For that reason, vaccination is important for the general population of adults and children, including IAC.

Further data on the potential clinical and economic impact of different protocols for MMRV vaccination are needed in order to establish the optimal immunization strategy in IAC.

The aim of this study is to evaluate the clinical and economic impact of different protocols of MMRV immunization of IAC. Developing a mathematical model, we compared three possible strategies in order to identify the most advantageous one and optimize MMRV vaccination protocol.

## 2. Materials and Methods 

### 2.1. Study Design

Features of IAC visiting the Meyer Children’s University Hospital, Florence (Italy), in the period 2009–2018 were included in the mathematical model. 

For each child included in the study, the following information was collected and entered in an electronic database, as previously described [1,2]. Briefly, the following information was retrieved for the purpose of the present study: country of origin, gender, age at first observation, vaccine documentation for MMRV, results of serological tests for measles, mumps, rubella, and varicella. At the first evaluation, all the children underwent a venepuncture and laboratory assessment including serologic tests.

All the other laboratory examinations were performed in the same laboratory at the Meyer Children’s University Hospital, using standardized techniques and according to manufacturers’ instructions.

### 2.2. Decision Analysis Model

The decision analysis model compared the strategy of vaccination based on documentation data versus presumptive vaccination or versus vaccination based on serotesting (Figure 1).

Immunization based on data reported in the vaccination documentation has been chosen as a comparator, since this is the one currently recommended by the Italian Ministry of Health. Therefore, this strategy was compared with: the strategy of pre-vaccination serotesting to all IAC, then, vaccinating only some children based on serological results; the strategy of presumptive vaccination of all IAC without performing serology tests. The model is referred to a hypothetical population of 1000 IAC per year.

Vaccination strategy based on documentation records (comparator) considers only reliable immunization documentation, and vaccination is performed in order to complete the immunization schedule appropriate for the age of the children. IAC with no dose recorded should receive two doses of vaccine. IAC who had already received one dose, should receive one additional dose of vaccine. The protected and unprotected IAC were calculated based on the assumed vaccination coverage and the vaccine antibody response rate. Finally, IAC who received two doses of vaccine in the country of origin, shouldn’t receive any dose, but IAC can be categorized as protected and unprotected based on serological test results. 

With regard to the pre-vaccination serology strategy (strategy 1), all IAC undergo a serology test for measles, mumps, rubella, and varicella. Therefore, based on sensitivity and specificity of the tests, IAC were classified as true positive, false positive, true negative and false negative. Children with negative results will receive the vaccination. Considering the true negative children, the protected and unprotected IAC were calculated based on the assumed vaccination compliance and the vaccine antibody response rate. Indeed, a proportion of the true negative IAC, despite immunization, will remain unprotected. False negative IAC are already protected and receiving the immunization will receive a booster effect. IAC with true positive and false positive results will not receive any vaccination, and while true positive IAC are indeed protected, false positive IAC not receiving any vaccinations, will remain unprotected.

In presumptive immunization of all IAC (strategy 2), all IAC should receive 2 doses of MMRV vaccine. Based on the assumed immunization coverage, all IAC will receive the first dose, while a small proportion of them will not return for the second dose. Thus, based on the vaccination coverage and the vaccine antibody response rate assumed, a number of children will be protected, while some others will not. Even performing the second dose, a percentage of unprotected children will persist. 

In all three strategies vaccination is always hypothesized using the quadrivalent MMRV vaccine. 

Therefore, for each strategy, based on the decision’s flow chart described above (Figure 1), IAC can be divided into protected (seropositive) and unprotected (seronegative). In fact, even if it seems that high levels of circulating antibodies are important for protection against outbreaks, current data do not give conclusive evidence as to what level of antibodies should be considered protective for each of the VPDs considered, IAC were defined protected when positivity of antibodies levels were serologically determinated and unprotected when negativity of antibodies levels were serologically determinated. 

Finally, avoided cases, defined as cases of illness that would have occurred if these children had not been vaccinated, are calculated multiplying the incidence rate for each disease for the number of protected cases. Moreover, possible cases of disease can be estimated by applying the incidence rate for each disease to the not protected cases. 

### 2.3. Population Data

The population data used in the mathematical model are derived from those of 1927 IAC referred to the Meyer Children’s University Hospital and published in 2020 [18]. The age-group distribution and features of this population were reported to be a hypothetical population of 1000 IAC (Table 1) based on a mathematical proportion. 

Table 2 shows the incidence of measles, mumps, rubella and varicella for each age group (<1 y, 1–4 y, 5–9 y, 10–14 y, 15–18 y) included in the model. The reported data is the average of the incidences in the period 2009-2018, and were obtained from the Tuscany Regional Health Agency (ARS) [13]. 

Vaccine documentation was assessed considering the number of doses recorded for MMRV. 

With regards to seroprevalence of antibody protection against MMRV, children were subdivided according to age in five groups (<1 y, 1–4 y, 5–9 y, 10–14 y, 15–18 y) and classified as protected or unprotected against each specific VPD on the basis of serotesting results. 

### 2.4. Vaccine Probabilities and Cost Estimates 

Since IAC have been reported to display a high coverage rate for vaccines [11], we assumed that 100% of IAC would receive the first MMRV dose, while 95% of IAC would complete the 2 dose series.

Vaccine MMRV antibody response was obtained from published data by the CDC [20,21,22,23] and was considered to be 98% for measles, 78% for mumps, 95% for rubella, and 65% for varicella for one dose; 99% for measles, 88% for mumps, 99% for rubella, and 95% for varicella for two doses.

Charges for serotesting (including cost of blood draw, antibody determination, laboratory technicians) were obtained from the price offered to the Meyer Hospital and the Department of Health Science, University of Florence, Florence (Italy). The analysis was developed applying the enzyme-linked immunosorbent assay (ELISA). Costs of the serology tests were calculated to be 6.28 Euro for measles per child, 5.86 Euro for mumps per child, 5.86 Euro for rubella per child, 6.28 Euro for varicella per child (Table 3).

The cost of the laboratory technician for each serology performed was calculated as follows. The hourly salary for a laboratory technician, which is 18.50 Euro, was multiplied by the number of hours necessary to set up the serology kit. The time required is on average six hours, during which the technician is engaged in selecting the sample, doing the dilution and analysing the results. The total is finally divided for the number of wells in the serological kit, that in our case is 46. In fact, in a 96-well plate, four wells were the control (positives and negatives) and the remaining wells, made of 46 reactive wells and 46 non-reactive wells, were used to perform the sample serology. This means that each serology sample had to be performed twice. 

Therefore, the cost of the laboratory technician for each well was calculated to be: 18.50 (Euro) × 6 (hours)/46 (wells) = 2.41 (Euro).

The charge for blood draw for the serology was considered to be 4.00 Euro.

The cost of vaccine was calculated including the charge of administering single immunization (5.91 Euro) (based on previously published Italian studies by Gasparini et al. and Di Pietro et al.) [24,25] and the cost of MMRV vaccine (obtained by the Tuscany Regional Healthy Agency). Two MMRV vaccines have been used during the study period and costs change over years, therefore the weighted average cost was calculated to be 46.30 Euro (Table 4).

In the analysis of each VPD, the cost of the vaccine was divided into the four parts of the MMRV vaccine. Even the cost of the blood draw was divided for four parts, as a single blood draw is enough to perform the four serologies. 

### 2.5. Sensitivity and Specificity of Serology Tests

Sensitivity and specificity of specific ELISA for MMRV are reported in Table 4. Since, additional serological methods are available, in the study further analyses were executed performing chemiluminescent immunoassay (CLIA) with sensitivity, specificity and costs reported in Table 4. Lastly, the costs of serologic tests are summarized in Table 3. 

In addition, as MMRV vaccine is performed after the first year of life, the analysis has been repeated excluding IAC <1 year of age.

### 2.6. Strategy Cost Estimates

The costs for each strategy were calculated for a population of 1000 IAC. The number of protected and unprotected IAC were used to calculate the number of avoided cases and the number of possible cases for each single disease and for all 4 VPDs together. This allows to directly compare the costs and the benefits of the strategies. 

Moreover, the comparator strategy was matched to the other two strategies by calculating the incremental cost-effectiveness ratio for protected children (ICER = costs difference / benefits difference).

### 2.7. Ethical Approval and Informed Consent

The study received approval by Meyer University Hospital Ethics Committee, the Ethical Code Number is 15/2010. 

Parents who consented to their adoptive children participating in the study have signed a written informed consent. 

## 3. Results

Vaccine documentation was assessed considering the number of doses recorded for MMRV. This data referred to the Meyer Children’s University Hospital and reported to a population of 1000 IAC, as entered in Table 5. To apply strategy 1, the number of vaccine doses needed to be administered was calculated based on this data.

On the basis of serotesting results, children were subdivided according to age in five groups and classified as protected or unprotected against each specific VPD as entered in Table 6. 

Compared with the immunization based on documentation strategy (comparator), presumptive immunization (strategy 2) for measles increases the cost per strategy from 17,445.68 Euro to 25,452.38 Euro. The number of protected IAC rises from 961 to 990; therefore, the cost per protected child increases from 18.16 Euro to 25.72 Euro. However, comparing the pre-vaccination serotesting strategies (strategy 1) with the comparator for measles, the cost per strategy increases less (from 17,445.68 Euro to 18,635.40 Euro), while the number of protected IAC increases more markedly from 961 to 994; therefore, the cost per protected child increases from 18.16 Euro to 18.75 Euro (Table 7). Therefore, as far as measles is concerned, the comparator is less expensive, but strategy 1 is more effective for number of protected IAC.

In fact, the ICER, defined above as the difference between the costs and the difference between the benefits (as number of protected IAC), was calculated comparing strategy 2 and strategy 1 with the comparator. By vaccinating all IAC, the ICER would be 279.28 Euro per protected child, when compared to the comparator. Indeed, comparing the strategy 1 vs the comparator, the ICER is 36.18 Euro per protected child (Table 8). This means that although the serology strategy is slightly more expensive, it is the most effective approach compared to the comparator and more advantageous compared to strategy 2. 

Comparing strategy 2 with the comparator for mumps, the cost per strategy increases from 9482.39 Euro to 25,452.38 Euro, whereas the number of protected IAC decreases from 915 to 875. The cost per protected child increases from 10.36 Euro to 29.09 Euro. Comparing strategy 1 with the comparator, the cost per strategy increases from 9482.39 to 19,453.99, while the number of protected IAC increases from 915 to 926. The cost per protected child increases from 10.36 Euro to 21.02 Euro, (Table 7). Therefore, the presumptive immunization strategy is dominated (more expensive and less effective) by comparator, while comparing strategy 1 to the comparator, the ICER is 944.77 Euro per protected child (Table 8).

Comparing strategy 1 with the comparator for rubella and varicella the cost of the strategies decreases from 18,245.20 Euro to 17,461.95 Euro and from 23,945.44 Euro to 21,390.65 Euro, respectively. Conversely, the number of protected IAC increases from 962 to 998 for rubella and from 930 to 966 for varicella. The cost per child protected decreases from 18.95 Euro to 17.49 Euro and from 25.74 Euro to 22.14 Euro, respectively. Therefore, the comparator is dominated by strategy 1 (less expensive and more effective). Moreover, for both VPDs, vaccination of all IAC is the most expensive strategy (25,452.38 Euro for rubella and for varicella). When compared to the comparator, strategy 2 increases the number of protected IAC from 963 to 995 for rubella and from 930 to 934 for varicella. (Table 7). Comparing strategy 2 to the comparator, the ICER is 222.82 Euro per protected child for rubella and 394.92 Euro per protected child for varicella (Table 8).

Finally, considering all VPDs together, comparing strategy 1 with the comparator, the cost of the strategy increases from 69,118.72 Euro to 76,941.99 Euro and even the number of protected IAC increases from 3769 to 3884. The cost per protected child increases from 18.34 Euro to 19.81 Euro. Comparing strategy 2 with the comparator, the cost of the strategy increases to a greater extent: from 69,118.72 Euro to 101,809.50 Euro and the number of protected IAC increases from 3769 to 3794. The cost per patient increases from 18.34 Euro to 26.84 Euro (Table 7). Therefore, compared to the other strategies, the comparator strategy is less expensive. On the other hand, the pre-vaccination serotesting strategy against MMRV together, improves the number of protected IAC with a minimum increment of costs compared with the immunization based on documentation strategy. Therefore, from a cost-effectiveness point of view, vaccination based on serotesting results in the most advantageous strategy. As a matter of fact, with an expenditure of 7823.28 Euro more than the comparator strategy, strategy 1 is associated with 115 more protected IAC (Table 7 and Table 8). 

Performing the analysis using CLIA, pre-vaccination serotesting is overall the most convenient approach (Table A1 and Table A2).

Finally, excluding children aged <1 year, performing both methods of serology (ELISA and CLIA), similar results were obtained (Table A3, Table A4, Table A5 and Table A6).

## 4. Discussion and Conclusions

Considering MMRV, pre-vaccination serotesting compared to immunization based on documentation in IAC improves outcomes with a minimal cost increase, while the presumptive immunization strategy is the most expensive and it is associated with a number of protected children lower than pre-vaccination serotesting. 

Although immunization based on documentation is the strategy currently recommended in Italy [26], in our analysis, the cost-saving of this strategy was minimal compared to the benefits of serotesting strategy, performing ELISA serology. On the other hand, all the other outcomes (total number of protected IAC, number of possible cases, and number of avoided cases) were greater than the pre-immunization serotesting approach. However, performing CLIA serologies, clinical and economic outcomes were more advantageous by applying the pre-vaccination serotesting strategy. In fact, in this case, pre-vaccination serotesting compared to the comparator decreases the costs, increases the percentage of protected IAC against all four VPDs, increases the number of avoided cases and decreases the number of possible cases. 

There is no standard criteria for measuring cost-effectiveness in vaccination strategies for IAC [11]. To our knowledge, this is the first clinical and economic analysis of the MMRV vaccine in IAC. Only Cohen et al. in 2006 conducted an economic analysis of pre-vaccination serotesting compared with presumptive immunization for poliomyelitis, diphtheria, and tetanus in IAC and immigrant children. This study demonstrated that the pre-vaccination serotesting for poliomyelitis increases the cost and decreases the percentage of patients protected [11]. Another study targeting child refugees analysed the most convenient strategy comparing pre-vaccination serotesting with presumptive immunization for varicella and concluded that pre-vaccination serotesting is cost-effective in children <5 years of age [8]. Other similar studies are available for hepatitis B and hepatitis A, but these refer to a population of native children [9,10]. 

Therefore, due to the shortage of published data, our study highlights the necessity of further studies investigating the most cost-effective immunization strategy that targets IAC.

Our study has several potential limitations. Firstly, the costs in our study are derived from the University of Florence (Health Science Department, Meyer Children Hospital) and may not reflect costs at other institutions. Secondly, no study investigated the coverage rates for the immunization in IAC, so we relied on expert opinions to evaluate the reported rates. Thirdly, a limited number of IAC underwent mumps serology testing in the Meyer Hospital population. Fourthly, in-directed costs (i.e., wages lost by caregivers) were not considered. Moreover, considering the strategy of the presumptive immunization of all IAC, the additional costs of potential side effects of unnecessary vaccination in protected children were not considered. However, severe adverse events are extremely rare, and we speculate that our results are not or are minimally affected by this omission. Lastly, there is no absolute association between being unprotected (seronegative) and not having protective antibody titers, since immune response to vaccines is complex and relies on different immunologic mechanisms which are not all explored by serology testing [27]. In fact, current data does not give conclusive evidence what level of antibodies should be considered protective for MMRV, even if it seems that high levels of circulating antibodies are important in protection against outbreaks [28,29,30]. Moreover, our analysis is based on several assumptions. Firstly, the vaccine selected is against MMRV, due to the fact that the monovalent vaccines are not used in medical practice in Italy. Secondly, the coverage rate was assumed to be 100% for the first dose of MMRV vaccine and 95% for the second dose. Indeed, adoptive parents are generally very attentive to their children’s medical needs and IAC are usually well integrated into the national health system. Moreover, in Tuscany (Italy), a first screening evaluation for newly arrived IAC is free of charge and a first blood draw is already performed in all IAC as a general assessment. Therefore, the costs for the serology could decrease, not considering the cost for the venepuncture. Finally, this model was developed by modifying input data of the methods used to perform the serology for measles, mumps, rubella and varicella (ELISA and CLIA). CLIA is the method currently used at the Anna Meyer Hospital of Florence. Varying the method for the serology tests, the sensitivity, specificity and costs, change. The obtained results are almost comparable to the results described above, except for the costs, and indeed in this case pre-vaccination serotesting is also the cost-saving strategy compared with the others. Similar results were obtained excluding children aged < 1 year.

Further studies are needed to assess the coverage rate of IAC to the immunization protocols. Moreover, it could be interesting to extend this analysis to the entire immigrant population. Probably the relation between documentation and serology could differ from IAC, and the assumed coverage rate could be lower. 

In conclusion, based on our analyses, considering MMRV vaccine, serotesting strategy appears to be the preferred option in IAC. However, results of this study refer to a particular kind of population: IAC often arrive to the new country with unknown or unreliable vaccination documentation. Moreover, IAC are subjected to numerous routine examinations promptly after their arrival that can be integrated with the MMRV serology. Therefore, the results of this study that highlight the importance of pre-vaccination serotesting can be applied only in IAC and cannot be extended to the general population. 

## Figures and Tables

**Figure 1 vaccines-08-00060-f001:**
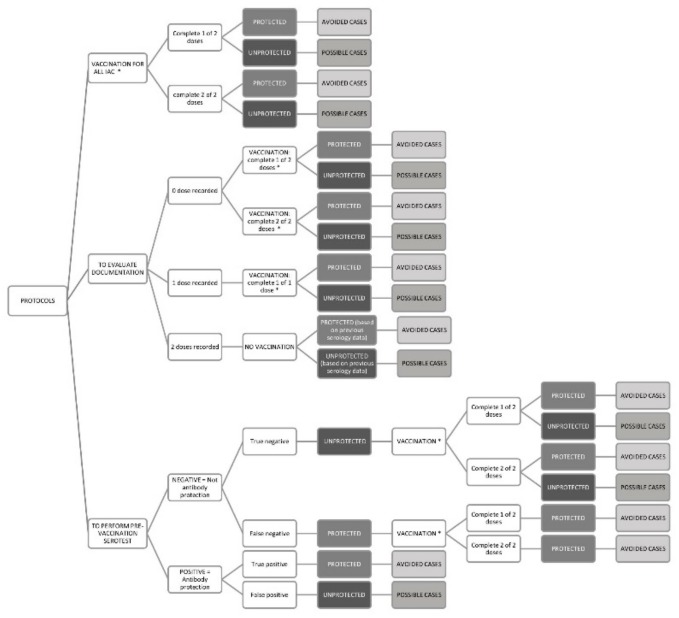
Decision analysis model comparing the strategy of vaccination based on documentation data against pre-vaccination serotesting or compared with presumptive vaccination with no testing. (* protected and unprotected IAC were calculated based on the compliance to the vaccination and the antibody response rate).

**Table 1 vaccines-08-00060-t001:** Study population according to age-groups and vaccine preventable disease reported to a population of 1000 internationally adopted children (IAC) per year.

Age Groups	Measles	Mumps	Rubella	Varicella	Source
<1 y	26	4	27	17	Meyer Hospital population reported to 1000 international adopted children per year (2009–2018)
1–4 y	391	373	390	338	
5–9 y	473	525	474	485	
10–14 y	96	94	95	124	
15–18 y	14	5	14	36	
TOTAL	1000	1000	1000	1000	

**Table 2 vaccines-08-00060-t002:** Measles, mumps, rubella and varicella incidences per 100,000 population per year according to age-groups (2009–2018) [19].

Age Groups	Measles	Mumps	Rubella	Varicella
<1 y	15.2	1.6	0.3	275.3
1–4 y	7.3	10.0	0.9	547.4
5–9 y	3.5	6.3	0.3	369.8
10–14 y	2.6	3.1	0.2	88.9
15–18 y	4.4	1.2	0.5	22.6

**Table 3 vaccines-08-00060-t003:** Cost estimates.

Costs	Base Cost Estimate, Euro		Sources
Cost of vaccine	46.30		Tuscany Regional Healthy Agency
Charge for administering immunization	5.91		[24,25]
Charge for blood draw for serology	4.00		Price
Cost of serology kit	ELISA	CLIA	Costs offered to the Department of Health Science, University of Florence, Italy / Meyer Children Hospital, University of Florence, Italy
-Measles	6.28	2.45	
-Mumps	5.86	2.45	
-Rubella	5.86	1.57	
-Varicella	6.28	1.96	
Charge for the laboratory technician per serology test	2.41		Meyer Children Hospital, University of Florence, Italy

**Table 4 vaccines-08-00060-t004:** Sensitivity and specificity of the serology tests.

Methods		Measles	Mumps	Rubella	Varicella
**ELISA**	Sensitivity	0.996	0.954	1	0.993
	Specificity	1	0.937	0.985	1
**CLIA**	Sensitivity	0.974	0.985	0.995	1
	Specificity	1	0.985	0.996	0.971

Note: information provided by the manufacturers.

**Table 5 vaccines-08-00060-t005:** Documentation of vaccination for the IAC enrolled from the Meyer Children’s University Hospital and referred to a hypothetical population of 1000 IAC.

Number of Doses Documented for IAC by Age Groups	<1 yrs	1–4 yrs	5–9 yrs	10–14 yrs	15–18 yrs
**Measles**					
0 doses	24	197	239	50	11
1 dose	2	170	132	18	2
2 doses (n seronegative)	0 (0)	24 (7)	103 (17)	28 (6)	2 (1)
**Mumps**					
0 doses	4	7	8	1	0
1 dose	0	339	306	39	4
2 doses (n seronegative)	0 (0)	27 (0)	211 (0)	53 (0)	1 (0)
**Rubella**					
0 doses	27	218	244	55	12
1 dose	1	158	138	17	1
2 doses (n seronegative)	0 (0)	13 (4)	92 (20)	24 (10)	1 (0)
**Varicella**					
0 doses	17	295	433	119	36
1 dose	0	36	40	3	0
2 doses (n seronegative)	0 (0)	6 (3)	13 (3)	2 (0)	0 (0)

**Table 6 vaccines-08-00060-t006:** Seroprevalence of the internationally adopted children (IAC) enrolled at the Meyer Children’s University Hospital and referred to a hypothetical population of 1000 IAC.

VPD	Measles		Mumps		Rubella		Varicella	
Seroprevalence of IAC by Age Groups	negatives	positives	negatives	positives	negatives	positives	negatives	positives
< 1 y	23	3	120	0	22	5	13	5
1–4 y	140	251	160	200	137	253	231	106
5–9 y	149	325	120	360	127	346	185	300
10–14 y	32	64	0	40	33	62	21	103
15–18 y	8	6	0	0	2	12	10	27

**Table 7 vaccines-08-00060-t007:** Clinical and economic outcomes per 1000 Internationally Adopted Children (IAC), by disease (ELISA method).

VPD	Strategy	Total Cost of the Strategy (Euro)	IAC ^1^ Protected (N)	Avoided Cases (N)	Possible Cases (N)	Cost Per Protected Children (Euro)
Measles	Comparator	17,445.68	960.83	0.0501	0.0018	18.16
	Strategy 1	18,635.40	993.72	0.0516	0.0003	18.75
	Strategy 2	25,452.38	989.50	0.0514	0.0005	25.72
Rubella	Comparator	18,245.20	962.78	0.00507	0.00013	18.95
	Strategy 1	17,461.95	998.46	0.00520	0.00001	17.49
	Strategy 2	25,452.38	995.13	0.00518	0.00003	25.58
Mumps	Comparator	9482.39	915.02	0.0664	0.0067	10.36
	Strategy 1	19,453.99	925.55	0.0641	0.0050	21.02
	Strategy 2	25,452.38	875.00	0.0639	0.0091	29.09
Varicella	Comparator	23,945.44	930.43	3.5382	0.2688	25.74
	Strategy 1	21,390.65	966.00	3.6623	0.1446	22.14
	Strategy 2	25,452.38	934.25	3.5566	0.2503	27.24
All VPDs together	Comparator	69,118.72	3769.06	3.6598	0.28	18.34
	Strategy 1	76,941.99	3883.72	3.7832	0.15	19.81
	Strategy 2	101,809.50	3793.88	3.6771	0.26	26.84

Note: IAC ^1^ = Internationally adopted children; comparator = vaccine based on vaccination records; strategy 1 = vaccine based on serology; strategy 2 = vaccine to all IAC.

**Table 8 vaccines-08-00060-t008:** Comparison of strategies according to disease (ELISA method).

VPD	Comparison of Strategies vs Comparator	Difference of Costs (Euro Per 1000 Children)	Difference in Protected Children (N Per 1000)	ICER (Euro Per Protected Case)
Measles	Serology	1189.72	32.89	36.18
	Presumptive immunization	8006.70	28.67	279.28
Rubella	Serology	−783.25	35.68	Dominant
	Presumptive immunization	7207.17	32.35	222.82
Mumps	Serology	9971.60	10.53	946.77
	Presumptive immunization	15,969.98	−40.02	Dominated
Varicella	Serology	−2554.79	35.56	Dominant
	Presumptive immunization	1506.93	3.82	394.92
All VPDs together	Serology	7823.28	114.66	68.23
	Presumptive immunization	32,690.78	24.81	1317.51

Note: Presumptive immunization of all internationally adopted children; comparator vaccinations of international adopted children according to their documentation of previous vaccinations; Serology = vaccinations of international adopted children according to serotesting results.

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
