# Peer review of "Assessment of the Clinical and Economic Impact of Different Immunization Protocols of Measles, Mumps, Rubella and Varicella in Internationally Adopted Children"

_vaccines, 2020, doi:10.3390/vaccines8010060_

Round 1
Reviewer 1 Report
The manuscript by Boccalini et al, describes a clinical and economic assessment of three different immunization protocols for measles, mumps, rubella and varicella in internationally adopted children in Italy. While this subject will likely be of interest to many readers and is worthy of publication, there are grammatical errors throughout. Additionally, the organizational structure/presentation must be improved prior to publication. It is strongly recommended that the authors seek review from a colleague well versed in the English language or from a professional editing service. Specific comments/suggestions:
(1) Lines 69-73 should be moved to the end of the introduction, as this describes the study objective.
(2) Line 92 is confusing as written. Did you mean to say "presumptive vaccination" instead of "pre-vaccination serotesting"?
(3) The Materials and Methods section would be significantly improved through significant re-organization. Instead of having one section titled "probabilities", authors should consider adding several more specific headings and break the sections up. For example, "2.3 Population Data" (Tables 1 & 2); "2.4 Vaccine Probabilities and Cost Estimates" (Tables 5 & 6); "2.5 Sensitivity and Specificity of Serology Tests" (Table 7).
(4) Lines 174-181; the text and table are redundant. Suggest that the authors choose one format (text or table).
(5) Data presented in Tables 3 & 4 is better suited for the results section; suggest describing this data at the beginning of the results.
(6) Table 7: "Sensibility" should read "Sensitivity". This typo is present throughout the table.
(7) Line 225; this should be numbered appropriately for the Methods and Materials section. Example, "2.6 Strategy Cost Estimates".
Author Response
Response to Reviewer 1 Comments
Thank you for your suggestions, we tried to improve the clarity of the manuscript following your observations.
Point 1: Lines 69-73 should be moved to the end of the introduction, as this describes the study objective.
Response: we have moved the study objective to the end of the introduction as suggested
Point 2: Line 92 is confusing as written. Did you mean to say "presumptive vaccination" instead of "pre-vaccination serotesting"?
Response: we have corrected typing error.
Point 3: The Materials and Methods section would be significantly improved through significant re-organization. Instead of having one section titled "probabilities", authors should consider adding several more specific headings and break the sections up. For example, "2.3 Population Data" (Tables 1 & 2); "2.4 Vaccine Probabilities and Cost Estimates" (Tables 5 & 6); "2.5 Sensitivity and Specificity of Serology Tests" (Table 7).
Response: we have broken the sections up adding more specific headings as you suggested
Point 4: Lines 174-181; the text and table are redundant. Suggest that the authors choose one format (text or table).
Response: we have deleted the table as suggested
Point 5: Data presented in Tables 3 & 4 is better suited for the results section; suggest describing this data at the beginning of the results.
Response: we moved Table 3 and 4 in the results section as suggested
Point 6: Table 7: "Sensibility" should read "Sensitivity". This typo is present throughout the table.
Response: we have replaced sensibility with sensitivity throughout the table as suggested
Point 7: Line 225; this should be numbered appropriately for the Methods and Materials section. Example, "2.6 Strategy Cost Estimates".
Response: we have re-numbered the heading appropriately
Reviewer 2 Report
Nice and straightforward paper with original and novel findings: this study deals with a interesting topic to the public health.
The authors provide relevant data but some modifications could increase the readability of the text.
In the Introduction the authors should complete the reasoning by reporting data about prevalence of MMRV in Italy and in other Countries. Previous studies have reported data about prevalence and have shown the importance of vaccination. Provide to address this aspect and to report some references.
In Materials and Methods the authors say: "Therefore, for each strategy, based on the decision’s tree steps described above, IAC can be divided 130 in protected (seropositive) and unprotected (seronegative)." No reference, not even in the introduction, is given for this statement.
It is challenging to support this claim from the literature. Current data do not give conclusive evidence what level of antibodies should be considered protective for each of the VPDs the authors have considered, even if seems that high levels of circulating antibodies are important in protection against outbreaks, as you have correctly stated albeit briefly in the Discussion (lines 348-351).
Address deeply this issue, supported by references,in the Introduction and/or Discussion.
Do the authors use the term "immunization" and "immunisation" indifferently or with different meanings? Provide for uniformity.
Author Response
Response to Reviewer 2 Comments
Thank you for your suggestions, we tried to improve the reliability of the text following your observations.
Point 1: In the Introduction the authors should complete the reasoning by reporting data about prevalence of MMRV in Italy and in other Countries. Previous studies have reported data about prevalence and have shown the importance of vaccination. Provide to address this aspect and to report some references.
Response: we implemented the introduction by reporting MMRV data in Italy and in other countries to explain the importance of vaccination also in the IAC population. We have reported these data with references.
Point 2: In Materials and Methods the authors say: "Therefore, for each strategy, based on the decision’s tree steps described above, IAC can be divided (line 130) in protected (seropositive) and unprotected (seronegative)." No reference, not even in the introduction, is given for this statement.
It is challenging to support this claim from the literature. Current data do not give conclusive evidence what level of antibodies should be considered protective for each of the VPDs the authors have considered, even if seems that high levels of circulating antibodies are important in protection against outbreaks, as you have correctly stated albeit briefly in the Discussion (lines 348-351).
Address deeply this issue, supported by references, in the Introduction and/or Discussion.
Response: we have improved this statement by supporting it with further references.
Point 3: Do the authors use the term "immunization" and "immunisation" indifferently or with different meanings? Provide for uniformity.
Response: we have standardized this term as suggested
Reviewer 3 Report
This is an interesting cost-effectiveness analysis of scenarios for vaccinating internationally adopted children in Italy. The approach is quite thorough and well explained and the presentation includes the essential elements for reporting economic evaluations.
Major comment
It seems No Ethical approval was obtained from Meyer Children’s University Hospital although vulnerable population was the subject of the study. A statement should be included.
Minor comments.
1. The rationale of why its important and to vaccinate or revaccinate international adopted children and why its currently under contention is well presented in the introduction. However, there is a need for additional information fairly outlining previous related studies.
2. Line 140. How the numbers in the age-groups numbers of the hypothetical population of 1,000 were generated needs to included in the text.
3. The tables e.g. Table 3, needs to reformatted for clarity.
Author Response
Response to Reviewer 3 Comments
Thank you for the appreciation of the work. We tried to improve the clarity of the manuscript following your observations.
Point 1: It seems No Ethical approval was obtained from Meyer Children’s University Hospital although vulnerable population was the subject of the study. A statement should be included.
Response: We obtained the Ethical approval and the consent by the adoptive parents for this study. We added a paragraph in “Material and methods” in order to specify it.
Point 2: The rationale of why it’s important and to vaccinate or revaccinate international adopted children and why its currently under contention is well presented in the introduction. However, there is a need for additional information fairly outlining previous related studies.
Response: The rationale of the study was implemented in the introduction. Economic studies in the literature dealing with this topic are few and all studies retrieved were mentioned in our study.
Point 3: Line 140. How the numbers in the age-groups numbers of the hypothetical population of 1,000 were generated needs to be included in the text.
Response: as suggested we specified that this hypothetical population of 1,000 was generated based on a mathematical proportion using Meyer Children’s University Hospital population.
Point 4: The tables e.g. Table 3, needs to be reformatted for clarity.
Response: we reformatted Table 3 as suggested
Round 2
Reviewer 1 Report
Thank you. All comments/suggestions have been satisfactorily addressed. Recommend publication.